# Prognosis Factors of Patients Undergoing Renal Replacement Therapy

**DOI:** 10.3390/jpm13040605

**Published:** 2023-03-30

**Authors:** José Manuel Muñoz-Terol, José L. Rocha, Pablo Castro-de la Nuez, Juan José Egea-Guerrero, Luis Gil-Sacaluga, Emilio García-Cabrera, Angel Vilches-Arenas

**Affiliations:** 1Department of Nephrology, Hospital Universitario Virgen del Rocío, 41013 Seville, Spain; 2Department of Medicine, University of Seville, 41009 Seville, Spain; 3Information System of the Autonomic Transplant Coordination of Andalusia (SICATA), 41013 Seville, Spain; 4Neurocritical Care Unit, Hospital Universitario Virgen del Rocío, 41013 Seville, Spain; 5Institute of Biomedicine of Seville (IBiS)/Consejo Superior de Investigaciones Científicas (CSIC), University of Seville, 41013 Seville, Spain; 6Preventive Medicine and Public Health Department, University of Seville, 41009 Seville, Spain; 7Department of Preventive Medicine, Hospital Universitario Virgen Macarena, 41009 Seville, Spain

**Keywords:** renal replacement therapy, end-stage kidney disease, kidney transplantation, premature mortality

## Abstract

Background: Survival in patients with end-stage kidney disease (ESKD) on renal replacement therapy (RRT) is less than that of the general population of the same age, and depends on patient factors, the medical care received, and the type of RRT used. The objective of this study is to analyze the factors associated with survival in patients undergoing RRT. Methods: We conducted a retrospective observational study of adult patients with an incident of ESKD on RRT in Andalusia from 1 January 2008 to 31 December 2018. Patient characteristics, nephrological care received, and survival from the beginning of RRT were evaluated. A survival model for the patient was developed according to the variables studied. Results: A total of 11,551 patients were included. Median survival was 6.8 years (95% CI (6.6; 7.0)). After starting RRT, survival at one year and five years was 88.7% (95% CI (88.1; 89.3)) and 59.4% (95% CI (58.4; 60.4)), respectively. Age, initial comorbidity, diabetic nephropathy, and a venous catheter were independent risk factors. However, non-urgent initiation of RRT and follow-up in consultations for more than six months had a protective effect. It was identified that renal transplantation (RT) was the most influential independent factor in patient survival, with a risk ratio of 0.13 (95% CI (0.11; 0.14)). Conclusions: The receiving of a kidney transplant was the most beneficial modifiable factor in the survival of incident patients on RRT. We consider that the mortality of the renal replacement treatment should be adjusted, taking into account both modifiable and nonmodifiable factors to achieve a more precise and comparable interpretation.

## 1. Introduction

Chronic kidney disease (CKD) has seen a significant increase in recent years, becoming one of the fastest growing noncommunicable diseases worldwide. According to the 2017 Global Burden of Disease Study, it is estimated that there are currently between 649.2 and 752.1 million individuals worldwide who are living with CKD in different stages [1]. Renal replacement therapy (RRT), whether by dialysis (hemodialysis (HD) or peritoneal dialysis (PD)) or renal transplantation (RT), can keep a patient with end-stage kidney disease (ESKD) alive. However, patients on RRT have an average loss of 9 potential years of life lost compared to patients without this treatment [2]. The European Renal Association-European Dialysis and Transplant Association (ERA-EDTA) registry performs annual survival analyses [3,4]. The results show a significant improvement in survival at one year and two years in patients with RRT when comparing the incident cohort in 2011–2015 with that in 2008–2012. This same fact had previously been established in this same registry with an increase of 12% in overall survival adjusted for age, sex, cause of ESKD, and country among incident patients with RRT between 1997 and 2006 [5]. Despite this trend of improvement in survival, it is insufficient, as it is lower compared to other disease survival trends such as cancer [6]. There is strong evidence that suggests that among the different forms of RT, offering RRT to patients with ESKD is the most beneficial option, as it not only results in lower mortality rates [7,8,9] and improved survival [10,11], but is also the most efficient treatment [12,13]. The main limitation of current studies is that survival data come from global case registries, with high statistical power in terms of the number of cases, but with a limited number of variables to adjust survival models, generally non-modifiable factors such as age, sex, and kidney disease. Additionally, RRT is not homogeneous across European countries [14], with this disparity in conservative treatment options, timing, frequency, and modality of onset of RRT depending on patient characteristics, which could affect European survival data [9]. Therefore, observational cohort studies that include more variables are necessary to establish modifiable risk factors, with the goal of improving the prognosis. Our objective was to study the survival of patients with incident ESKD in RRT in Andalusia and to identify the modifiable or nonmodifiable factors involved in patient survival.

## 2. Materials and Methods

We conducted a retrospective cohort study that covered the period from 2008 to 2019 and included all centers located in the Andalusian region of southern Spain. The study population consisted of 8.4 million individuals. We included all patients who started RRT during the study period and had at least one year of follow-up. Patients included on 31 December 2018 were followed up until 31 December 2019 or until an event occurred. However, we excluded pediatric patients, those with cardiorenal syndrome, those receiving RRT for conditions other than ESKD, and those who initiated RRT outside of Andalusia.

Information on RRT patients was obtained from the SICATA basic CKD module [15]. SICATA is a mandatory population-based registry maintained by the Andalusian Autonomous Transplant Coordination Office. It captures data on patients who have begun RRT, including renal transplantation (RT), peritoneal dialysis (PD), or hemodialysis (HD). All patients who started RRT in Andalusian hospitals and dialysis centers were required to sign an informed consent document, which granted permission for the inclusion of their data in SICATA and for their information to be used in epidemiological and research studies.

The study was carried out according to the ethical principles outlined in the Declaration of Helsinki (Fortaleza-Brazil Review, October 2013) [16] and was approved by the Ethics Committee of the Virgen del Rocio-Macarena Hospitals on 26 January 2021. All records included the following variables: sex, employment status, age at the beginning of RRT, adjusted Charlson index for CKD [17], programmed onset of RRT, etiology of ESKD [18], time to follow-up of ESKD by nephrologists, modality of RRT initiation, vascular access used for the first RRT, previous kidney transplant, serology for hepatitis B virus (HBV), serology for hepatitis C virus (HCV), and serology for human immunodeficiency virus (HIV).

A descriptive analysis of all cases was carried out using proportions and percentages of qualitative variables. For quantitative variables, the mean and standard deviation or median and quartiles (P25; P75) were calculated, depending on whether they followed a normal distribution, and confidence intervals were calculated at 95% (CI95%). Subsequently, a subgroup analysis was performed according to the patient’s vital status and whether they had received RT. The relationship between qualitative variables was studied using the chi-squared test, and for dichotomous qualitative and quantitative variables, Student’s *t*-test was used. In case the normality requirement is not met (Kolmogorov or Shapiro–Wilk test, depending on the size of the subgroups), the Mann–Whitney U test was performed. In case of significant differences, the confidence intervals were determined to quantify these differences at 95%. The comparison of numerical variables between more than two groups was carried out using the ANOVA test or the Kruskal–Wallis test.

The survival time of all patients with RRT was estimated using the Kaplan–Meier method, and the median survival and quartiles (P25; P75) were calculated. A graphic representation of the survival curve and tables with survival percentages at one year, three years, five years, and ten years were made. Univariate survival analysis was performed using Kaplan–Meier curves, in which the dependent variable was attempted to be predicted from the independent variables. Log-Rank, Brelow, or Tarone–Ware tests were used to compare the equality of survival time distributions between groups. Finally, Cox regression was used to create survival-time models until the event occurred, including hypothetical predictor variables (covariables), categorical and continuous. A multivariate Cox proportional hazards regression model was constructed once the application requirements were validated with variables with a significance level of 0.15 in the univariate study. Hazard ratios (HR) and 95% ICs were calculated for the variables selected by the model after validation of the application requirements. The predictive capacity of the patient’s vital state considering mortality from all causes was estimated by building a receiver operating characteristic (ROC) curve with the patient’s survival probability and calculating the area under the curve. In all tests, a significance level of 0.05 was considered. Data analysis was carried out using the SPSS statistical package version 26.

## 3. Results

We conducted a retrospective cohort that included 11,551 patients with ESKD who were undergoing RRT. The leading cause of ESKD in our study population was diabetic nephropathy at 24.5% CI95% (23.7%; 25.3%). Comorbidity, measured by the Charlson index at the beginning of replacement therapy, was 6.0 points CI95% (5.9; 6.1 points), and diabetes was the most common comorbidity diagnosed in 38.5% CI95% (37.6%; 39.4%) of the patients. The median age of the patients at the start of RRT was 65 years CI95% (65; 66 years), in which 67.8% of the patients CI95% (66.9%; 68.6%) started RRT on a programmed schedule and 73.3% CI95% (72.5; 74.1) had a nephrologist follow-up for 6 months or more prior to starting RRT. The characteristics of our study cohort population are presented in Table 1.

During the follow-up period, 3776 patients or 32.7% CI95% (31.8%; 33.5%) received renal transplantation (RT) as RRT, and 413 (10.9%) began replacement therapy directly through RT. The characteristics of the patients depending on whether they received RT are shown in the Appendix A Appendix A.

After one year of RRT starting, the survival rate was 88.7% CI95% (88.1%; 89.3%), after 5 years was 59.4% CI95% (58.4%; 60.4%), and at 10 years was 37.4% CI95% (36.0%; 38.8%) (Table 2). The median survival time after the start of RRT was 6.8 years (6.6; 7.0) (Figure 1). At the end of the follow-up, 4900 patients did not survive (42.2% CI95%) (41.5; 43.3). The characteristics of the surviving patients at the end of follow-up are shown in Table 3. Depending on the modality of RRT initiation, we found different survival rates: patients who start with RT have a higher survival rate and patients who start hemodialysis have the lowest survival rate. (Figure 2).

In univariate Cox regression, we identified some nonmodifiable factors associated with prognosis: age-onset RRT, sex, comorbidities, diabetic nephropathy, and HCV-positive serology; as well as modifiable factors related to RRT: time of follow-up by the nephrologist, programmed onset of RRT, vascular access devices, and modality of RRT at start. All variables associated with prognosis are detailed in Table 4. Finally, in the Cox regression model, we independently found two nonmodifiable factors associated with survival: Charlson index score (HR 1.15 CI95%) (1.14; 1.16) and male sex (HR 1.08 CI95%) (1.02; 1.15); and four factors related to RRT: not scheduled start (HR 1.08 CI95%) (1.01; 1.16), venous catheter (HR 1.46 CI95%) (1.03; 2.07); follow-up for 6 months or longer by the nephrologist (HR 0.92 CI95%) (0.86; 0.99), and renal transplantation (HR 0.13 CI95%) (0.11; 0.14) (Table 5). The recipient of a kidney transplant, at any time during RRT, was revealed to be the most protective factor in our study, with an 87% reduction in the risk of mortality. (Figure 3). The area under the curve, which estimates the discrimination capacity of the predictive model, was 0.784 CI95% (0.775; 0.792), *p* < 0.001 (Appendix A, Appendix A).

## 4. Discussion

In our study, five years after starting renal replacement therapy, the survival rate is 59.4% (58.4–60.4). Studies in European series show a survival rate of 51.1% (51.0–51.2) [4], whereas the Spanish registry reports a rate of 57% (CI95% (56.6–57.4)) [19]. Variations in these results can be attributed to the fact that the survival data being compared are not adjusted and do not consider factors such as sex, age, comorbidities, patient follow-up, or the proportion of patients who undergo transplantation in relation to the overall population receiving renal replacement treatment.

In our cohort, the median age at which RRT was onset was 65 years, 62.8% were men, and 24.5% of the onset of RRT had diabetic nephropathy; in the data recorded in the European cohort, they were younger (the median age was 63 years) and there was an equal proportion of men and a lower rate of diabetic nephropathy (20%). On the contrary, in the Spanish cohort, they were older (median age of 67.2 years) and with a higher percentage of men (65.4%) and the same rate of diabetic nephropathy (24.8%). This variability of population characteristics is partly responsible for the variability of unadjusted survival, since in all series, the age of onset and male sex and diabetic nephropathy are reasons for the adjusting mortality rates. However, the difference is that in our study, we adjusted the age with the Charlson index adjusted for ESKD along with other comorbidities. This has also been reported in other studies in which other cardiovascular, neoplastic, and psychiatric comorbidities, included in Charlson, are independently associated with survival [20]. Furthermore, in the study by Beddhud et al. [16], comorbidity had a better ability to predict the evolution of patients with RRT than diabetes alone. Therefore, we consider that more data on comorbidity should be provided in records to establish a better survival adjustment and not limit it only to the adjustment for diabetic nephropathy. The adjusted Charlson index for ESKD presents the best adjustment with survival, but limited survival prediction capacity alone [21], and is a simple way to control the effect of adjacent diseases in patients who begin RRT, especially considering the average ages of onset, where the presence of more than one pathology is very frequent.

Despite progress in survival in RRT [9], it is still unclear when individuals with end-stage renal disease (ESRD) should transition to it for optimal outcomes, considering factors such as age, comorbidities, and race/ethnicity, to ensure the best possible survival [22]. Early dialysis initiation is not supported by evidence [23], and conservative management shows better results in elderly patients with multimorbidity [24]. This evidence is consistent with our results that patients with 6 months of follow-up prior to starting RRT have shown better survival results.

Vascular access and type of central venous catheter have been identified in multiple studies as mortality risk factors for patients with RRT [25,26,27], as well as in our study. This method of vascular access has complicated the comparison of survival rates between different dialysis techniques, as seen in a study conducted in the Canary Community by Garcia-Canton [28]. The study found that patients on peritoneal dialysis and hemodialysis with internal vascular access had comparable survival rates; however, those on hemodialysis with catheter-based vascular access had lower survival rates compared to those on peritoneal dialysis.

The most important factor that affects survival is the type of renal replacement therapy (RRT) chosen. Transplantation has been shown to be the best option in all studies, both European and national [4,19]. Furthermore, the proportion of patients who receive transplants is a significant factor, with an average of 30% in all groups of patients who begin RRT. However, this percentage can vary from 29% in the Spanish cohort to 38% in the European cohort, depending on organ availability and patient eligibility criteria. Over the years, there has been a shift in the characteristics of a candidate, particularly concerning age. As healthcare providers acquire more expertise in treating elderly patients and those who were previously deemed unsuitable for transplantation, such as individuals over 75 years old [29], In addition, there has been a trend towards postponing the initiation of RRT and employing a conservative management approach with vigilant monitoring. This change in approach has been associated with better prognosis [30,31]. Our study also supports this finding, as scheduled initiation and follow-up by a nephrologist for more than 6 months were found to be independent factors associated with better patient survival, also reported by other authors [32,33].

The main limitation of our work is that we conducted a retrospective study. All patients included have presented at the time of their inclusion an indication of starting RRT uninterruptedly, but the criteria may not have been completely uniform among the different treatment centers and may have varied over the period of the study. There is very little information prior to inclusion in RRT; the comorbidity of patients has been collected only at the beginning of RRT and has not been recorded, neither the severity nor if new comorbidities have appeared during their evolution in RRT; and some classic risk factors in the general population and in patients with CKD are not registered. The functional status of the patients has also not been included in the parameters collected. The only aspect that has collected information in this area has been the employment situation of the patient, the completion of which does not exceed 25% and which could not be included in multivariate statistical analysis for this reason. The absence of analytical data at the beginning of the RRT and during follow-up has also been complete. There are no data that allow assessing whether the different dialysis techniques have been optimal or whether the factors related to the correct functioning of the kidney transplant have been the most appropriate.

Additionally, our study has significant advantages. The inclusion of a high number of patients with a large number of events in their evolution allowed us to conduct a thorough multivariate Cox analysis, including all desired variables, which allowed us to determine the impact of these variables on patient survival during RRT. Our model has been able to correctly classify almost 80% of patients as alive or deceased. The temporal period of the study has been recent; in addition, it has been carried out in a homogeneous geographical area, i.e., Andalusia, and, at the same time, with a large population base of more than 8 million people. The number of variables that have been studied is very high and cover various prognostic aspects (sociodemographic and clinical variables related to nephrological care, comorbidities, and mortality) and with a high completion of the different variables, greater than 99% in most of them. This last aspect has differentiated it very significantly from other studies that have presented much lower completion, such as the Vonesh study [34], which does not have information on comorbidities in up to 45% of patients. The inclusion of cases has been complete, without the relevant loss of incident cases in RRT.

## 5. Conclusions

The receiving of a kidney transplant was the most beneficial modifiable factor in the survival of incident patients on RRT. We consider that mortality should be adjusted for the patient’s comorbidities and not just for diabetes (non-modifiable factors), and initiate renal replacement therapy should be scheduled with adequate follow-up by the nephrologist, avoiding central catheters, and with a kidney transplant whenever possible.

## Figures and Tables

**Figure 1 jpm-13-00605-f001:**
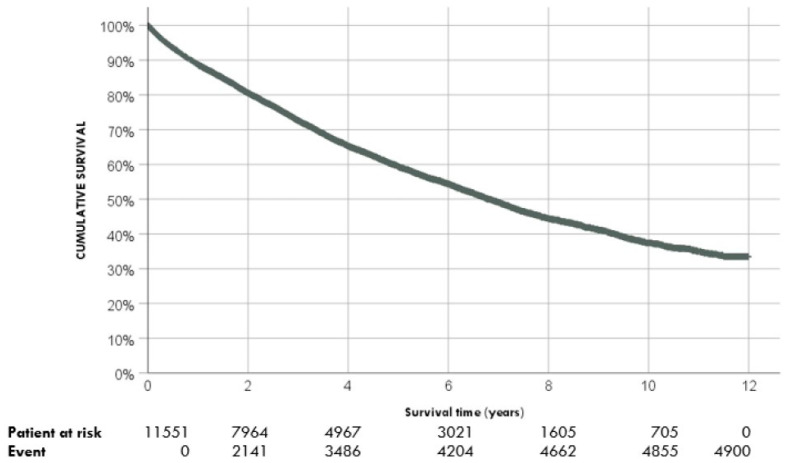
Global survival of renal replacement therapy.

**Figure 2 jpm-13-00605-f002:**
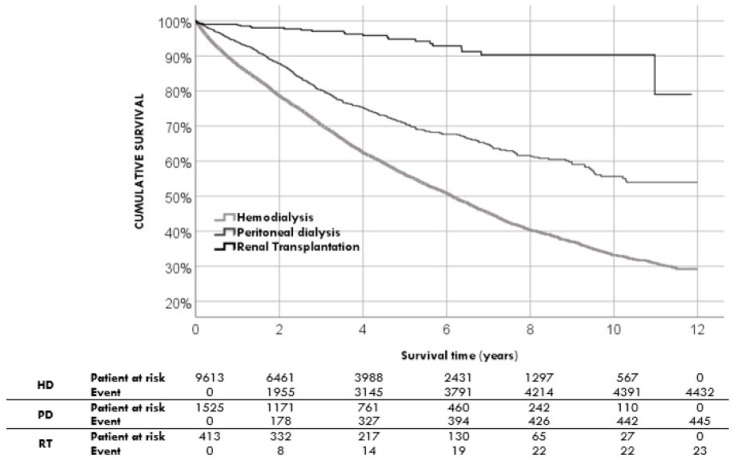
Univariate survival analysis comparing the modality of renal replacement therapy at the start.

**Figure 3 jpm-13-00605-f003:**
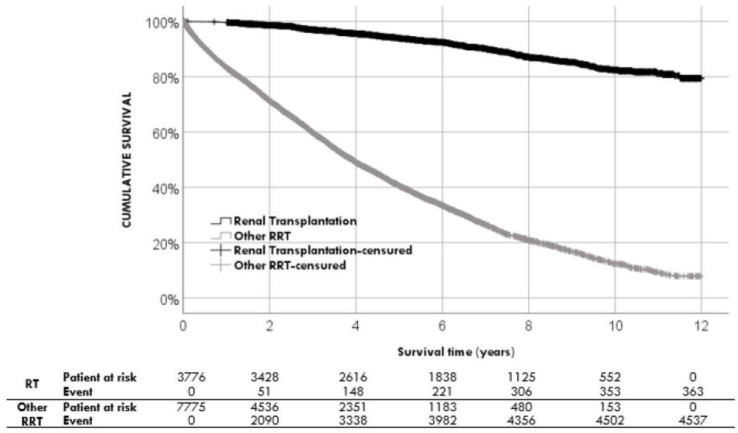
Cox multivariate analysis comparing patients with renal transplantation.

**Table 1 jpm-13-00605-t001:** Demographic, clinical, and follow-up characteristics of the patients included in the cohort.

	*N =* 11,551 *(%)*	CI95%
Sex		
Male	7254 (62.8)	(61.9; 63.7)
Female	4293 (37.2)	(36.3; 38.1)
Occupational status *		
Active	2414 (20.9)	(19.8; 21.9)
Inactive	4677 (79.1)	(78.1; 80.2)
Comorbidities		
Median Charlson index score (IQR)	6 (2)	(6; 7)
Diabetes mellitus	4443 (38.5)	(37.6; 39.4)
Congestive heart failure	2184 (18.9)	(18.2; 19.6)
Myocardial infarction	2066 (17.9)	(17.2; 18.6)
Peripheral vascular disease	1737 (15.0)	(14.4; 15.7)
COPD	1406 (12.2)	(11.6; 12.8)
Solid tumour localised	1031 (8.9)	(8.4; 9.5)
Cerebrovascular accident	983 (8.5)	(8.0; 9.0)
Connective tissue disease	476 (4.1)	(3.8; 4.5)
Peptic ulcer disease	372 (3.2)	(2.9; 3.6)
Mild liver disease	388 (3.4)	(3.0; 3.7)
Moderate to severe liver disease	195 (1.7)	(1.5; 1.9)
Leukemia/lymphoma	175 (1.5)	(1.3; 1.8)
Dementia	148 (1.3)	(1.1; 1.5)
Metastatic solid tumour	117 (1.0)	(0.8; 1.2)
HIV	96 (0.8)	(0.7; 1.0)
AIDS	23 (0.2)	(0.1; 0.3)
HBV	139 (1.2)	(1.0; 1.4)
HCV	371 (3.2)	(2.9; 3.5)
End-stage kidney disease aetiology		
Diabetic nephropathies	2832 (24.5)	(23.7; 25.3)
Glomerular disease	1747 (15.1)	(14.5; 15.8)
Hypertension and renovascular disease	1684 (14.6)	(13.9; 15.2)
Renal tubulo-interstitial nephropathy	1222 (10.6)	(10.0; 11.1)
Familial/hereditary nephropathies	971 (8.4)	(7.9; 8.9)
Systemic diseases affecting the kidney	389 (3.4)	(3.1; 3.7)
Miscellaneous renal disorders	305 (2.6)	(2.4; 2.9)
Chronic renal failure; aetiology uncertain	2401 (20.8)	(20.1; 21.5)
End-stage kidney disease follow-up		
Without previous follow-up	1541 (13.3)	(12.7; 14.0)
ESKD ≤ 6 months of follow-up	1543 (13.4)	(12.7; 14.0)
ESKD > 6 months of follow-up	8467 (73.3)	(72.5; 74.1)
Renal replacement therapy starts		
Median age-onset RRT	65	(65.0; 66.0)
Programed	7754 (67.8)	(66.9; 68.6)
Not programmed.	3683 (32.2)	(31.4; 33.1)
Renal replacement therapy modalities at the beginning		
Hemodialysis	9613 (83.2)	(82.5; 83.9)
Peritoneal dialysis	1525 (13.2)	(12.6; 13.8)
Renal transplantation	413 (3.6)	(3.3; 3.9)
Vascular access devices at start		
Venous catheter	5227 (45.2)	(44.8; 46.0)
Arteriovenous (AV) fistula + AV graft	4510 (39.0)	(38.2; 39.8)
None	1814 (15.7)	(15.0; 16.4)
Transplantation Waiting List situation		
Permanently excluded	6358 (55.0)	(54.1; 55.9)
Temporary excluded	423 (3.7)	(3.3; 4.0)
Screening not completed	994 (8.6)	(8.1; 9.1)
Included and waiting	229 (2.0)	(1.7; 2.2)
Included and transplanted	3547 (30.7)	(29.9; 31.5)
Vital Status		
Alive	6651 (57.6)	(56.7; 58.5)
Death	4900 (42.2)	(41.5; 43.3)
Age at death (median)	75	(74.7; 75.4)
Death causes		
Infectious	1118 (22.8)	(21.6; 23.9)
Cardiac	983 (20.1)	(19.0; 21.2)
Neoplasm	582 (11.9)	(11.0; 12.8)
Vascular	501 (10.2)	(9.3; 11.0)
Gastrointestinal	154 (3.1)	(2.6; 3.6)
Hepatic	42 (0.9)	(0.6; 1.1)
Accidental	10 (0.2)	(0.1; 0.3)
Multiple causes	519 (10.6)	(9.7; 11.5)
Undetermined	992 (20.2)	(19.1; 21.3)

* 4460 (38.6%) missing data.

**Table 2 jpm-13-00605-t002:** Survival of incident patient in RRT.

Survival	%	CI95%
1 year	88.7	(88.1; 89.3)
3 years	72.6	(71.8; 73.4)
5 years	59.4	(58.4; 60.4)
10 years	37.4	(36.0; 38.8)

**Table 3 jpm-13-00605-t003:** Patient characteristic assessment related to survival.

	Alive *N* = 6651		Death *N* = 4900		*p*-Value
	*N* (%)	CI95%	*N* (%)	CI95%	
Sex					
Male	2701 (40.6)	(39.4; 41.8)	4552 (92.9)	(92.2; 93.6)	0.001
Female	3945 (59.4)	(58.2; 60.6)	348 (7.1)	(6.4; 7.8)	
Occupational Status					
Active	1811 (64.4)	(62.6; 66.2)	603 (14.1)	(13.0; 15.1)	<0.001
Inactive	1003 (35.6)	(33.8; 37.3)	3674 (85.9)	(84.9; 86.9)	
Comorbidities					
Median Charlson index score	5.0	(5.0; 5.1)	7.3	(7.2; 7.3)	<0.001
Diabetes mellitus	2113 (31.8)	(30.7; 32.9)	2330 (47.6)	(46.2; 49.0)	<0.001
Congestive heart failure	789 (11.9)	(11.1; 12.7)	1395 (28.5)	(27.2; 29.7)	<0.001
Myocardial infarction	800 (12.0)	(11.3; 12.8)	1266 (25.8)	(24.6; 27.1)	<0.001
Peripheral vascular disease	637 (9.6)	(8.9; 10.3)	1100 (24.4)	(21.3; 23.6)	<0.001
COPD	578 (8.7)	(8.0; 9.4)	828 (16.9)	(15.9; 18.0)	<0.001
Solid tumour localised	469 (7.1)	(6.5; 7.7)	562 (11.5)	(10.6; 12.4)	<0.001
Cerebrovascular accident	452 (6.8)	(6.2; 7.4)	531 (10.8)	(10.0; 11.7)	<0.001
Connective tissue disease	269 (4.0)	(3.6; 4.5)	207 (4.2)	(3.7; 4.8)	0.181
Peptic ulcer disease	152 (2.3)	(1.9; 2.7)	220 (4.5)	(3.9; 5.1)	<0.001
Mild liver disease	181 (2.7)	(2.4; 3.1)	207 (4.2)	(3.7; 4.8)	<0.001
Moderate to severe liver disease	73 (1.1)	(0.9; 1.4)	122 (2.5)	(2.1; 3.0)	0.004
Leukemia/lymphoma	63 (0.9)	(0.7; 1.2)	112 (2.3)	(1.9; 2.7)	<0.001
Dementia	44 (0.7)	(0.5; 0.9)	104 (2.1)	(1.7; 2.6)	<0.001
Metastatic solid tumour	36 (0.5)	(0.4; 0.7)	81 (1.7)	(1.3; 2.0)	<0.001
HIV	52 (0.8)	(0.6; 1.0)	44 (0.9)	(0.7; 1.2)	0.497
AIDS	12 (0.2)	(0.1; 0.3)	11 (0.2)	(0.1; 0.4)	0.117
HBV	88 (1.3)	(1.1; 1.6)	51 (1.0)	(0.8; 1.4)	0.169
HCV	179 (2.7)	(2.3; 3.1)	192 (3.9)	(3.4; 4.5)	<0.001
End-stage kidney disease aetiology					<0.001
Diabetic nephropathies	1397 (21.0)	(20.0; 22.0)	1435 (29.3)	(28.0; 30.6)	
Glomerular disease	1302 (19.6)	(18.6; 20.5)	445 (9.1)	(8.3; 9.9)	
Hypertension and renovascular disease	819 (12.3)	(11.5; 13.1)	865 (17.7)	(16.6; 18.7)	
Renal tubulo-interstitial nephropathy	609 (9.2)	(8.5; 9.9)	613 (12.5)	(11.6; 13.4)	
Familial/hereditary nephropathies	775 (11.7)	(10.9; 12.4)	196 (4.0)	(3.5; 4.6)	
Systemic diseases affecting the kidney	151 (2.3)	(1.9; 2.6)	238 (4.9)	(4.3; 5.5)	
End-stage kidney disease follow-up					
Without previous follow-up	833 (12.5)	(11.7; 13.3)	708 (14.4)	(13.5; 15.5)	0.002
ESKD ≤ 6 months of follow-up	863 (13.0)	(12.2; 13.8)	680 (13.9)	(12.9; 14.9)	
ESKD > 6 months of follow-up	4955 (74.5)	(73.4; 75.5)	3512 (71.7)	(70.4; 72.9)	
Renal replacement therapy at start					
Median age onset RRT	57.5	(57.1; 57.8)	69.7	(69.4; 70.1)	<0.001
Programed	4825 (73.0)	(71.9; 74.1)	2929 (60.7)	(59.3; 62.0)	<0.001
Not programmed.	1784 (27.0)	(25.9; 28.1)	1899 (39.3)	(38.0; 40.7)	
Renal replacement therapy modalities at the beginning					
Hemodialysis	5181 (77.9)	(76.9; 78.9)	4432 (90.4)	(89.6; 91.2)	<0.001
Peritoneal dialysis	1080 (16.2)	(15.4; 17.1)	445 (9.1)	(8.3; 9.9)	
Renal transplantation	390 (5.9)	(5.8; 6.0)	23 (0.4)	(0.3; 0.6)	
Vascular access devices at start					
Venous catheter	2548 (38.3)	(37.6; 38.9)	2679 (54.7)	(53.5; 56.9)	<0.001
Arteriovenous (AV) fistula + AV graft	2724 (41.0)	(40.2; 41.8)	1786 (36.5)	(35.2; 37.8)	
None	1379 (20.7)	(19.8; 21.7)	435 (8.9)	(8.1; 9.7)	
Transplantation					
Transplantation	3413 (51.3)	(50.1; 52.5)	363 (7.4)	(6.7; 8.2)	<0.001

**Table 4 jpm-13-00605-t004:** Univariate Cox evaluation of prognostic factors in incident RRT patients.

	N	HR	CI95%	*p*-Value
Age-onset RRT	11,551	1.06	(1.06; 1.06)	<0.001
Sex				0.008
Female	4293	1		
Male	7258	1.08	(1.02; 1.15)	
Diabetic nephropathies				<0.001
No	8719	1		
Yes	2832	1.45	(1.37; 1.55)	
End-stage kidney disease follow-up				<0.001
ESKD > 6 months of follow-up	8647	1		
ESKD ≤ 6 months of follow-up	3084	1.24	(1.17; 1.32)	
Renal replacement therapy at start				<0.001
Programed	7754	1		
Not programmed.	3683	1.67	(1.58; 1.77)	
Vascular access devices at start				<0.001
None	1814	1		
AV fistula + AV graft	4510	1.64	(1.48; 1.82)	
Venous catheter	5227	2.69	(2.43; 2.97)	
Renal replacement therapy modalities at the beginning				<0.001
Renal transplantation	413	1		
Peritoneal dialysis	1525	5.42	(3.57; 8.24)	
Hemodialysis	9613	9.67	(6.42; 14.57)	
Renal transplantation				<0.001
Yes	3776	1		
No	7775	12.58	(11.29; 14.02)	
HCV serology				<0.001
Negative	11,180	1		
Positive	371	1.37	(1.18; 1.58)	
Comorbidities				
Myocardial infarction	2066	2.1	(2.0; 2.2)	<0.001
Congestive heart failure	2184	2.4	(2.2; 2.5)	<0.001
Peripheral vascular disease	1737	2.1	(1.9; 2.2)	<0.001
Dementia	148	2.4	(2.0; 2.9)	<0.001
COPD	1406	1.8	(1.7; 2.0)	<0.001
Peptic ulcer disease	372	1.5	(1.3; 1.7)	<0.001
Mild liver disease	388	1.5	(1.3; 1.7)	<0.001
Solid tumour localised	1031	1.7	(1.5; 1.8)	<0.001
Leukemia/lymphoma	175	2.7	(2.3; 3.3)	<0.001
Diabetes mellitus	4443	1.8	(1.7; 1.9)	<0.001
Cerebrovascular accident	983	1.7	(1.5; 1.8)	<0.001
Moderate to severe liver disease	195	2.1	(1.7; 2.5)	<0.001
Metastatic solid tumour	117	3.0	(2.4; 3.8)	<0.001
Charlson index score				<0.001
Low (2–4)	3749	1		
Medium (5–6)	2881	4.2	(3.8; 4.6)	
High (7–8)	2764	6.6	(5.9; 7.3)	
Very high (>8)	2157	9.9	(9.0; 10.9)	

**Table 5 jpm-13-00605-t005:** Multivariate Cox regression model of prognostic factors in RRT patients in incident.

	HR	CI95%	*p*-Value
Charlson index score	1.15	1.14; 1.16	<0.001
Sex male	1.08	1.02; 1.15	0.007
Diabetic nephropathies	1.16	1.08; 1.16	<0.001
ESKD > 6 months of follow-up	0.92	0.86; 0.99	0.020
RRT start not programed	1.08	1.01; 1.16	0.025
Vascular access devices at start			
None	1	Ref.	<0.001
Arteriovenous (AV) fistula + AV graft	1.12	0.79; 1.59	
Venous catheter	1.46	1.03; 2.07	
Renal transplantation	0.13	0.11; 0.14	<0.001

## Data Availability

The data presented in this study are available on request from the corresponding author. The data are not publicly available due to privacy restrictions.

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
