# Peer review of "Prognosis Factors of Patients Undergoing Renal Replacement Therapy"

_jpm, 2023, doi:10.3390/jpm13040605_

Round 1

Reviewer 1 Report (New Reviewer)

In this retrospective cohort study, the authors analyze factors associated with survival in patients undergoing kidney replacement therapy. The study included 11,551 patients and had a number of variables which authors managed to control for to estimate survival rates. Overall, the methodological aspect of the study is rigorous enough.

My remarks are as follows;

-The justification for conducting this study is not solid enough. Aren’t the factors associated with survival in this population known already?

-In the discussion, there is need to highlight why non-urgent initiation of KRT and follow-up in consultations for more than six months had a protective effect.

-Suggest using the term ‘Renal replacement therapy/treatment’ rather than, ‘Replacement renal treatment’.

-In the last paragraph of the introduction, suggest using the term ‘non-modifiable’ rather than ‘not modifiable’.

-May the authors confirm that the SICATA registry contains data even for patients who are not transplant candidates.  

-The authors report that all patients sign an informed consent for their data to be collected. Would there be a proportion of patients who choose not to sign the consent form?

-What are the implications of this study to clinical practice?

Author Response

Reviewer 1

In this retrospective cohort study, the authors analyze factors associated with survival in patients undergoing kidney replacement therapy. The study included 11,551 patients and had a number of variables which authors managed to control for to estimate survival rates. Overall, the methodological aspect of the study is rigorous enough.

Response:

On behalf of all authors of this manuscript, I want to express my true gratitude to the reviewer for your time and dedication to review our manuscripts. In the next paragraph we respond to all your accurate and precise comments that clearly improve our work. Thank you again for your time, dedication, and expertise.

My remarks are as follows;

  1. The justification for conducting this study is not solid enough. Aren’t the factors associated with survival in this population known already?

Response: The justification of the article is focused on the need to identify modifiable variables in the treatment of patients undergoing renal replacement therapy to improve their survival. Although large survival series have identified variables associated with the survival of these patients, these variables are non-modifiable, which makes it difficult to improve patient management. Furthermore, the disparity in the treatment of these patients is highlighted at the European level and its possible implications on patient survival. Therefore, we have included a paragraph in the Introduction in which we emphasize that the treatment of patients undergoing renal replacement therapy is not homogeneous among European countries and this may have implications for patient survival.

-In the discussion, there is need to highlight why non-urgent initiation of KRT and follow-up in consultations for more than six months had a protective effect.

Response: Thanks for your comment, we add a paragraph including the importance about the follow-up for six moths during consultation. Currently, there is no strong evidence of the best time to start RRT in patients with ESKD, and we think our result adds knowledge on this controversial topic.

-Suggest using the term ‘Renal replacement therapy/treatment’ rather than, ‘Replacement renal treatment’.

Response: We change all terms replacement renal treatment/therapy by renal replacement treatment/therapy. Thanks for your suggestion.

-In the last paragraph of the introduction, suggest using the term ‘non-modifiable’ rather than ‘not modifiable’.

Response: We change not modifiable by non-modifiable in the last paragraph of the introduction; tanks again for your suggestion.

-May the authors confirm that the SICATA registry contains data even for patients who are not transplant candidates. 

Response: Yes, we confirm that SICATA is responsible for recording data from patients with chronic kidney disease (CKD) and monitoring the progress of treatment of hemodialysis and peritoneal dialysis. The reviewer could check it on the SICATA website (In Spanish)

https://www.sspa.juntadeandalucia.es/servicioandaluzdesalud/ayudadigital/aplicaciones/asistenciales/sicata-registro-de-transplantes

-The authors report that all patients sign an informed consent for their data to be collected. Would there be a proportion of patients who choose not to sign the consent form?

Response: Thank you for the question, but I cannot provide you any information about patients who have not signed informed consent because we only have information for those who have signed it.

-What are the implications of this study to clinical practice?

Response: Thank you again for the question, Based on the current patient profile with an average age of 65 years, multiple comorbidities, and non-modifiable variables such as age, gender, comorbidity, and underlying disease, clinicians may consider exploring other options to improve the survival of patients initiating renal replacement therapy. For example, if feasible, delaying the initiation of renal replacement therapy over time may be advantageous and avoid the use of venous catheter. However, since this is a retrospective study, we cannot draw definite conclusions. Further studies are required to assess whether prolonging the onset of therapy improves patients' survival prospects.

Reviewer 2 Report (Previous Reviewer 2)

Muñoz-Terol JM, et al demonstrated a study about factors associated with survival in RRT patients in Andalusia, 2008-2018. Authors showed that the survival rate was 89 and 59% at 1 and 5 years, respectively. Authors also showed that age, comorbidity, DN, and venous catheter were risk factors, while non-urgent initiation of RRT, consultation >6 months, and, particularly, RT were protective factors for mortality.

The manuscript was already well-written. I have some minor comments.

Minor issues

-        Please change N=11.551 (%) to N=11,551 (%) in table 1 in both sections

-        Please use similar abbreviation for the entire manuscript

o   RT vs TR: line 53, line 93, line 131

o   HIV vs VIH: table 3, table 1s

o   CI vs IC: line 111, line 174

o   Table 1s vs s1: line 259

-        Please change 4900 in line 136 to 4,900

-        Please check the format of references no 2, 12, 18, 28, 29

Author Response

Reviewer 2

Muñoz-Terol JM, et al demonstrated a study about factors associated with survival in RRT patients in Andalusia, 2008-2018. Authors showed that the survival rate was 89 and 59% at 1 and 5 years, respectively. Authors also showed that age, comorbidity, DN, and venous catheter were risk factors, while non-urgent initiation of RRT, consultation >6 months, and, particularly, RT were protective factors for mortality.

On behalf of all authors of this manuscript, I would like to express my sincere gratitude to the reviewer for taking the time to carefully review and provide valuable feedback on my work.

The manuscript was already well-written. I have some minor comments.

Minor issues

-        Please change N=11.551 (%) to N=11,551 (%) in table 1 in both sections

Response: We change it, thank you for your review.

-        Please use similar abbreviation for the entire manuscript

o   RT vs TR: line 53, line 93, line 131

o   HIV vs VIH: table 3, table 1s

o   CI vs IC: line 111, line 174

o   Table 1s vs s1: line 259

Response: We change it, thank you for your correction.

-        Please change 4900 in line 136 to 4,900

Response: We change it, thank you again for your review.

-        Please check the format of references no 2, 12, 18, 28, 29

Response: We check all reference carefully and change it, thank you again for your corrections.

This manuscript is a resubmission of an earlier submission. The following is a list of the peer review reports and author responses from that submission.

Round 1

Reviewer 1 Report

This manuscript analyze the factors associated with survival in patients undergoing renal replacement therapy. Authors conclude that The receiving of a kidney transplant was the most beneficial modifiable factor in the survival of incident patients on RRT.

Major comments:

Please elaborate more about RRT in a separate section, when it is needed, selection of patients and how it is affected by comorbidities.

Please explain how it affect ESRD, and stage 5 CKD.

It is not very clear how RRT affect the extent of patients undergoes transplantation. Does patients receiving RRT survive more and need transplantation at later stages.Please represent data with more graphical way, and show a correlation between patients receiving vs not receiving RRT and extent of transplantation.

How does age/ and comorbidies affect the RRT and time of transplantation? Please elaborate in discussion part.

Minor suggestions: Please make a abbreviations table for all mentioned abbreviation in the manuscript.

Author Response

REVIEW 1

This manuscript analyze the factors associated with survival in patients undergoing renal replacement therapy. Authors conclude that The receiving of a kidney transplant was the most beneficial modifiable factor in the survival of incident patients on RRT.

We are deeply grateful to the reviewer for taking the time to provide feedback on our work. Their thoughtful and constructive comments have helped us improve and refine our manuscript, and I am truly appreciative of their efforts. We respond to the above comments.

Major comments:

Q1. Please elaborate more about RRT in a separate section, when it is needed, selection of patients and how it is affected by comorbidities. Please explain how it affect ESRD, and stage 5 CKD.

Response

Thank you very much for your contribution. We have included a new section in the materials and methods where all information about the stage of initiation of renal replacement therapy is described. In our study, all patients who initiated renal replacement therapy had end-stage renal disease. Patients with stage 5 CKD were followed up in pre-dialysis consultations until the moment of initiation, and it was at the time of initiation when they were included in the study.

Q2. It is not very clear how RRT affect the extent of patients undergoes transplantation. Does patients receiving RRT survive more and need transplantation at later stages. Please represent data with more graphical way and show a correlation between patients receiving vs not receiving RRT and extent of transplantation.

Response.

All included patients received RRT, of which 9613 (83.2%) started HD, 1525 (13.2%) on PD, and 413 (3.6%) directly started kidney transplantation. Data shown in Table 1. And figure 2. Patients who initiated HD and PD, a total of 3363 eventually had kidney transplantation. All patients in our study are in TRS; therefore, we cannot present data on TRS vs. non-TRS. However, we can evaluate the comorbidity and age of transplanted patients compared to those who did not in Table 1 of the supplementary material attached to the manuscript.

Q3. How does age/ and comorbidies affect the RRT and time of transplantation? Please elaborate in discussion part.

Response.

Thank you for your feedback. We have included a discussion in our text (lines 180-199) regarding the impact of age and comorbidities on RRT survival. This finding is unique compared to other studies where comorbidities are only adjusted for the disease that caused the RRT, such as diabetic nephropathy. In contrast, we adjusted for all comorbidities, measured by the Charlson score index. Additionally, the age of onset for RRT was consistently around 60-65 years across all the series. Our group has also published a retrospective study in the Journal of Clinical Medicine (2022 Dec 21;12(1):51) that quantified the number of years of potential life lost (YPLL) in dialysis versus renal transplantation.

Q4. Minor suggestions: Please make an abbreviations table for all mentioned abbreviation in the manuscript.

Response.

We have included a table in the supplemental materials that contains all the abbreviations mentioned in the text. Appreciated your comment.

Author Response

REVIEW 2

Muñoz-Terol JM, et al demonstrated a study about factors associated with survival in RRT patients in Andalusia, 2008-2018. Authors showed that the survival rate was 89 and 59% at 1 and 5 years, respectively. Authors also showed that age, comorbidity, DN, and venous catheter were risk factors, while non-urgent initiation of RRT, consultation >6 months, and, particularly, RT were protective factors for mortality.

The manuscript was quite well-written. I have some questions and comments as follow.

Thank you for your thoughtful comment. We make it a point to respond to each comment individually.

Major issues

Q1.  What did the authors mean by “none” for “vascular access devices at start dialysis”? Does this mean PD?

Response.

It means that both patients receiving peritoneal dialysis and those undergoing renal transplantation directly are included in the category “none vascular access”.

Q2.  Why did authors decide not to include patients who followed up duration of less than 1 year? Does this make the most severe participants not to include, and bias the survival?

Response.

We apologize to the reviewer because the wording is confusing. What we meant to say is not that patients with less than one year of follow-up were excluded, but that all included patients had the possibility of at least one year of follow-up. Patients included on December 31, 2018, were followed up until December 31, 2019, or until an event occurred. We change the text at lin 68-69.

Minor issues

Q3.  - Table 1

o I may suggest authors to combine table 1 and table 3 together, and get rid of all CI 95% columns.
o For Charlson comorbidity index, please show median (IQR), not 95% CI
o There’s no need to show 95% CI in the baseline table.
o Authors may show only male for sex, active for occupation status, and programmed, since the counterpart is the rest of participants that readers would know spontaneously.

Response

We have tried to combine both tables, but it is impossible for us to do so. The importance of including the confidence interval in the total sample is to quantify that value in the reference population, which we believe is essential to be able to compare ourselves with other cohorts. Therefore, we do not see the need to eliminate it. We have included the dispersion value (IQR) in the Charlson data, but we also believe that considering the confidence interval value of the Charlson median is necessary for the reasons previously stated.

Q4.  - Table 5, please make sure if HR and 95% CI of diabetic nephropathy is correct.

- Please make sure to use similar abbreviation for the entire manuscript, RT vs TR, HIV vs VIH, CI vs IC, etc.

- Please make sure to use . not , for the decimal place.

Response We have carefully reviewed the text again and corrected all errors. We appreciate your patience and thorough review.

Q5.  - Please add the number of participants at risk under the KM curves.

Response After considering your comment, we have made changes to the KM-curves by including the patient at risk and the number of events. We believe that these modifications have improved the clarity of the curves. Thank you for your valuable input.

Round 2

Reviewer 1 Report

none.